# Improving Optoelectrical Properties of PEDOT: PSS by Organic Additive and Acid Treatment

**Shui-Yang Lien [1,2,3], Po-Chen Lin [4], Wen-Ray Chen [5,*], Chuan-Hsi Liu [6], Po-Wen Sze [7], Na-Fu Wang [8] and Chien-Jung Huang [4,*]**

[1] School of Opto-Electronic and Communication Engineering, Xiamen University of Technology, Xiamen 361024, China; sylien@xmut.edu.cn
[2] Department of Materials Science and Engineering, Da-Yeh University, Changhua 51591, Taiwan
[3] Fujian Key Laboratory of Optoelectronic Technology and Devices, Xiamen University of Technology, Xiamen 361024, China
[4] Department of Applied Physics, National University of Kaohsiung, Kaohsiung University Rd., Kaohsiung 81148, Taiwan; m1094307@mail.nuk.edu.tw
[5] Department of Electronic Engineering, National Formosa University, Wenhua Rd., Yunlin 632301, Taiwan
[6] Department of Mechatronic Engineering, National Taiwan Normal University, Heping East Rd., Taipei 10610, Taiwan; liuch@ntnu.edu.tw
[7] Department of Electrical Engineering, Kao Yuan University, Zhongshan Rd., Kaohsiung 82151, Taiwan; t20029@cc.kyu.edu.tw
[8] Center for Environmental Toxin and Emerging-Contaminant Research, Super Micro Mass Research & Technology Center, Department of Electronic Engineering, Cheng Shiu University, Chengcing Rd., Kaohsiung 82146, Taiwan; nafuwang@gcloud.csu.edu.tw
* Correspondence: chenwr@nfu.edu.tw (W.-R.C.); chien@nuk.edu.tw (C.-J.H.)

**Abstract:** This article demonstrates the change of structural and optical properties of poly (3,4-ethylene dioxythiophene): polystyrene sulfonate (PEDOT: PSS) by organic additive and acid treatment. The addition of sorbitol and maltitol can disperse the micelles of PEDOT: PSS. The mechanism of the bond-breaking reaction was investigated and a model for the bond-breaking reaction is also proposed. Furthermore, multiple formic acid treatments were found to reduce the PSS content of PEDOT: PSS, resulting in an enhancement in conductivity ($4.2 \times 10^4$ S/m).

**Keywords:** PEDOT: PSS; organic additive; acid treatment; haze; sheet resistance

## 1. Introduction

With the increasing popularity of smartphones, tablet computers, terminal devices, and portable game consoles, the demand for flexible, inexpensive, and lightweight touchscreen panels has steadily increased. Currently, indium tin oxide (ITO) is the most common material for touchscreen manufacture, but there are some drawbacks for ITO, such as the shortage of it on the Earth and requirements of high temperature and a vacuum process for fabrication. Overall, an increasing number of studies have focused on discovering a replacement material for ITO.

The advantages of poly (3,4-ethylenedioxythiophene): polystyrene-sulfonate (PEDOT: PSS) include transparency, low haze, and large-area application; hence, this material is favorable for optoelectrical devices and replaces ITO [1–3]. However, the potential of PEDOT: PSS to replace ITO in touchscreens has met with limited success because PEDOT: PSS exhibited low conductivity, which is a fundamental drawback. Therefore, more and more studies have focused on how to enhance conductivity. In 2002, Kim et al. investigated the effect of enhanced conductivity from added dimethyl sulfoxide (DMSO), N, N-dimethyl formamide (DMF), and tetrahydrofuran (THF) in PEDOT: PSS [4]. After that, more and more teams added polar organic additives (DMSO, EG, DMF, sorbitol, meso-erythritol, and xylitol) to PEDOT: PSS films to improve conductivity [5–7]. In addition,

the thermoelectric performance of PEDOT: PSS has been improved by adding graphene materials in recent research [8]. Recently, Sandeep Sharma et al. presented samples of PEDOT: PSS embedded with different reduced graphene oxide concentrations to improve conductivity and transmittance [9]. Apsar Pasha and Syed Khasim reported the effect of poly (ethylene glycol) treatment and silver nanoparticle distribution in PEDOT: PSS films, which has enhanced the conductivity of PEDOT: PSS films from 2 to 420.33 S/cm [10]. The conductivity of PEDOT: PSS film can not only be enhanced by additives, but also by annealing and acid treatment [11–16]. It has been reported that the conductivity of PEDOT: PSS film can be increased by annealing [11]. PEDOT films covered by insulating PSS are modified to remove some of the limiting insulating PSS layers during thermal treatment, but the haze value in PEDOT: PSS film increases [12]. For acid treatment, J.Y. Kim et al. also pointed out that the phenomenon of the Coulomb interaction between the PEDOT and the PSS is reduced by formic acid (FA) treatment [4]. Besides this, FA treatment could also be utilized to remove PSS from the PEDOT: PSS film, resulting in its higher conductivity [13,14]. After FA treatment, the PEDOT: PSS film was still highly transparent. Recently, Syed Khasim et al. reported high conductivity (1826 S/cm) and high transmittance (87%) of PEDOT: PSS film via camphor sulphonic acid [15]. The above research indicates that the conductivity and transmittance of PEDOT: PSS film can be improved by additive and acid treatment. It shows that PEDOT: PSS has great potential for use in optoelectrical devices, such as solar cells, touch panels, and organic devices [16–18].

In this study, organic additives such as sorbitol and maltitol were added to decrease the haze value and FA treatment was used to increase conductivity. Additionally, a detailed investigation of the haze and conductivity of PEDOT: PSS after the addition of organic additives and FA treatment is presented.

## 2. Materials and Methods

### 2.1. Material

The PEDOT: PSS was purchased from Heraeus (1.3–1.7 wt% from H. C. Starck Baytron P AI-4083, purchased from Heraeus Co., Hanau, Germany), maltitol was purchased from Alfa Aesar (Alfa Aesar, Shanghai, China, $C_{12}H_{24}O_{11}$, 97%), sorbitol was purchased from Sigma (MO, St. Louis, USA, $C_6H_{14}O_6$, 98%), and formic acid was purchased from Sigma (MO, St. Louis, USA, $CH_2O_2$, 98%).

### 2.2. Experimental Details

The glass substrate was cut to 3 cm × 3 cm and cleaned with acetone, methanol, and deionized water in an ultra-sonicator for 10 min before using oxygen plasma for surface passivation at 20 W for 120 s. Sorbitol (4 wt%) and maltitol (4 wt%) were doped in PEDOT: PSS, and stirred with a stirrer at 300 rpm and 100 °C for 20 min, as shown in Figure 1a. PEDOT: PSS was coated onto the substrate by using a spinner at 500 rpm for 10 s and 2000 rpm for 20 s. Subsequently, the film was baked at 100 °C for 20 min on a hot plate, as shown in Figure 1b. The FA was coated onto the PEDOT: PSS film using a spinner at 500 rpm for 10 s and 2000 rpm for 20 s. Next, the film was baked at 100 °C for 15 min on a hot plate, as shown in Figure 1c. The film was immersed in FA for 20 min. Afterward, the film was baked at 100 °C for 10 min on the hot plate. Then, the film was secondarily immersed in FA for 20 min. Finally, the film was baked at 100 °C for 10 min, as shown in Figure 1d.

### 2.3. Characteristic Measurements

The surface topography of the film was observed by Scanning Electron Microscopy (SEM; JEOL-6330TF; Taiwan). The haze of the thin film was investigated using a haze meter (Cat. No. AT-4725; USA). The sheet resistance of the film was measured using a 4-point Probe (JANDEL RM3000; UK). The uniformity of the film was analyzed by atomic force microscopy (AFM; XE-100; Korea). The composition of the bonds in PEDOT: PSS without and with organic additives were characterized by Fourier-transform infrared (FTIR;

PerkinElmer Spectrum RXI; Malaysia) and nuclear magnetic resonance spectroscopy (NMR; Mercury Plus 300 MHz; USA).

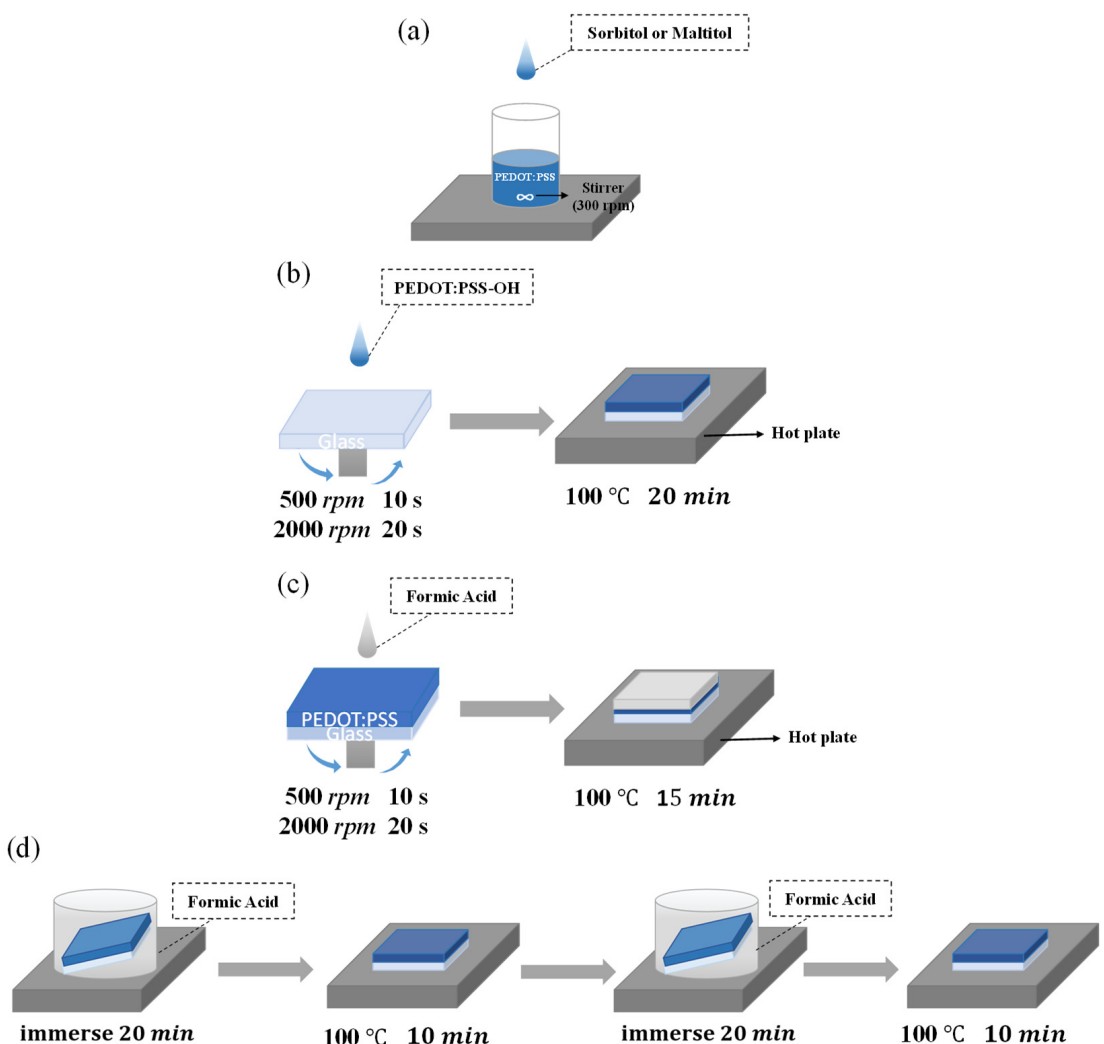

**Figure 1.** (**a**) Organic additive was doped in PEDOT: PSS. (**b**) The PEDOT: PSS was coated on the glass substrate and baked on a hot plate at 100 °C for 20 min. (**c**) The FA was coated on the PEDOT: PSS film, which was named acid treatment. (**d**) The PEDOT: PSS film was immersed in FA for about 20 min, which was named multi-acid treatment.

## 3. Results

### 3.1. External Factor of Haze

For industrial processes, the spin-coating process would generally be recommended at speeds of more than 1000 rpm to maintain the best uniformity [19]. However, it is possible for a nanoscale order to obtain a uniform film by spin coating at low speed (500~600 rpm) [20,21]. There is an article indicating that PEDOT: PSS is also a nanoscale order, with the result that it can obtain a uniform film using a low pre-coating speed [22]. In addition, one of studies points out that using a low-speed coating before a high-speed coating can also significantly reduce the roughness of the film [23]. Therefore, the method, which is a two-stage coating, can not only evenly spread the solution but also effectively improve the evenness of the film. The evenness of the film is related to the haze of the film. The relationship between spin-coating speed and the uniformity of film is presented in Table 1. The evenness of PEDOT: PSS film spun by various spin-coating speeds was measured by AFM, as shown in Figure 2. It can be seen in Figure 2a,b that the lower the pre-coating speed is, the less surface undulation the film has. This phenomenon illustrates

that lower pre-coating speeds can evenly spread the PEDOT: PSS on the entire substrate. This is consistent with the results of Ning, H. et al. [24]. Table 1 displays the root mean square deviation ($R_q$) of 1.53 nm and a haze of 0.43% for the higher pre-coating speed for 1000 rpm. Additionally, an $R_q$ of 1.26 nm and haze of 0.25% can be observed for a lower pre-coating speed of 500 rpm. According to the above results, the more uniform the film is, the lower the haze value is, owing to less diffuse reflection [25].

**Table 1.** Transmittance, haze, sheet resistance, conductivity, and $R_q$ of different pre-coating speeds for PEDOT: PSS films.

| Spin Ratio (rpm) | Transmittance (%) | Haze (%) | Sheet Resistance (Ω/sq) | Conductivity (S/m) | $R_q$ |
|---|---|---|---|---|---|
| 500 (first stage) 2000 (second stage) | 88.2 | 0.25 | $6.1 \times 10^5$ | 6.4 | 1.26 |
| 1000 (first stage) 2000 (second stage) | 86.6 | 0.43 | $5.9 \times 10^5$ | 6.6 | 1.53 |

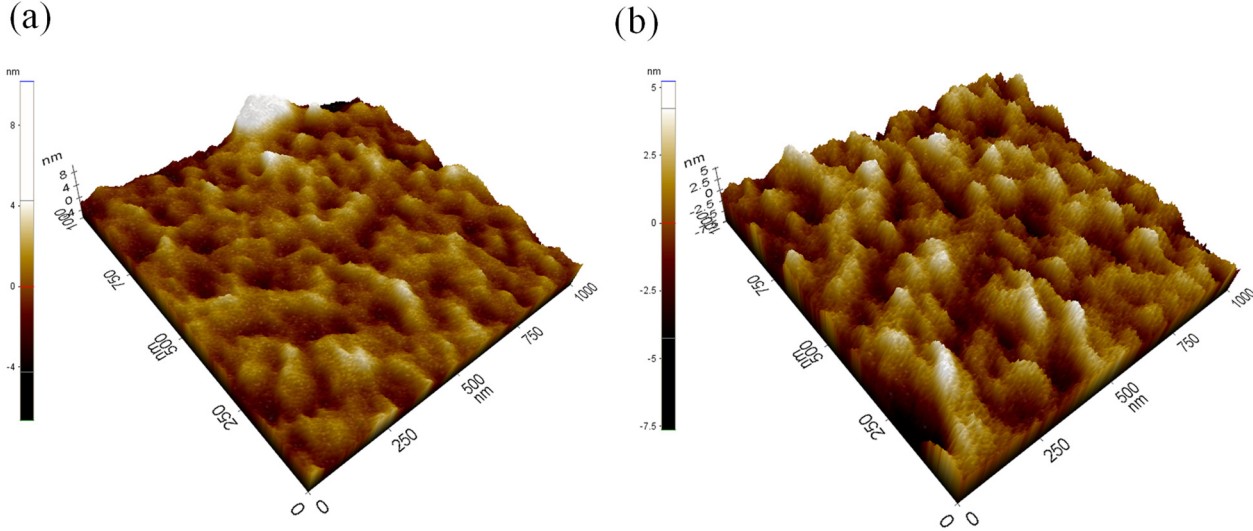

(a)   (b)

**Figure 2.** Three-dimensional AFM images of PEDOT: PSS at different pre-coating speeds of (**a**) 500 rpm and (**b**) 1000 rpm.

*3.2. Organic Additive Doping in PEDOT: PSS*

A schematic of pristine and added organic additives in PEDOT: PSS is shown in Figure 3. In Figure 3a, there are hydroxy sulfite bonds between individual PEDOT: PSS particles and hydrogen bonds develop between $HSO_3$ groups of the PSS shell of individual gel particles. With PEDOT: PSS doped with sorbitol and maltitol, the hydroxyl group reacts with hydroxyl sulfite bonds to generate sulfite ions and water. The PEDOT: PSS particles are dispersed in water after the bond-breaking reaction, as shown in Figure 3b. This phenomenon illustrates that the chain interactions between the PEDOT: PSS particles can be decreased by the organic additive. The smaller the chain interaction is, the larger the distance between PEDOT: PSS particles is. The transparency of the film can be improved by a larger distance between PEDOT: PSS particles [26]. Once the concentration of sorbitol exceeds 4 wt%, the haze value is increased, as shown in Table 2. This is due to the fact that the solubility of sorbitol in PEDOT: PSS has exceeded the upper limit. Thus, the haze value will not change, i.e., optimization.

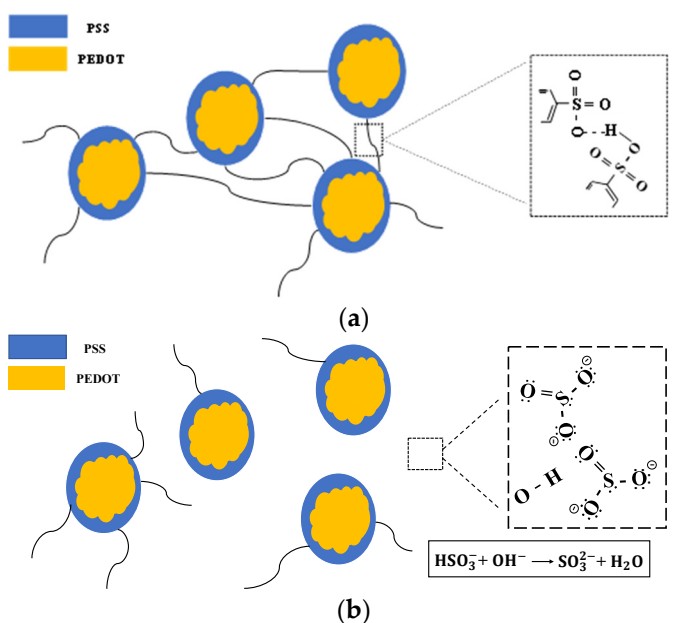

**Figure 3.** (**a**) A schematic of the hydrogen bond between hydrogen sulfite groups of the PEDOT: PSS. (**b**) A schematic of the hydroxyl group reaction with hydroxyl sulfite to produce sulfite ions and water after doping with organic additives.

**Table 2.** Transmittance and haze of PEDOT: PSS doped with various percentage concentrations (wt%) of sorbitol and maltitol.

| Sample | Sorbitol | | Maltitol | |
|---|---|---|---|---|
| Concentration (wt%) | Transmittance (%) | Haze (%) | Transmittance (%) | Haze (%) |
| pristine | 87.5 | 0.31 | 87.5 | 0.31 |
| 2 | 87.6 | 0.16 | 85.4 | 0.45 |
| 4 | 90.3 | 0.14 | 86.2 | 0.41 |
| 6 | 88.3 | 0.18 | 86.0 | 0.47 |
| 8 | 87.9 | 0.20 | 85.1 | 0.53 |

From Figure 4a and Table 3, it can be observed that the $R_q$ and the value of haze of pristine PEDOT: PSS film are 1.21 nm and 0.31%, respectively. In Figure 4b, the surface roughness of PEDOT: PSS-sorbitol film is more uniform than that of pristine film. The PEDOT: PSS with sorbitol of 4 wt% (hereinafter referred to as PEDOT: PSS-sorbitol) causes the value of $R_q$ to decrease and the haze value to be reduced, as shown in Table 3. Adding sorbitol to PEDOT: PSS can not only change the surface evenness but also disturb the interchain between PEDOT: PSS particles, with the result that the haze value and $R_q$ are down to 0.14% and 0.8 nm, respectively. Besides this, the PEDOT: PSS-sorbitol can also enhance the conductivity four-fold due to interchain interaction. As shown in Figure 4c, the surface of the PEDOT: PSS film with maltitol of 4 wt% (hereinafter referred to as PEDOT: PSS-maltitol) has more undulation than that of pristine film. As shown in Table 3, the $R_q$ of the PEDOT: PSS-maltitol has increased a little to 1.34 nm due to phase separation between PSS and PEDOT. The enrichment of PEDOT accumulates on the film after PSS separation, enhancing the conductivity of PEDOT: PSS-maltitol film. The addition of maltitol can improve conductivity but the haze value is increased. As shown in Figure 4d, the PEDOT: PSS film with a mixture of sorbitol and maltitol in a ratio of 3:1 (hereinafter referred to as PEDOT: PSS-sorbitol-maltitol) has a more uniform surface than the pristine film. Therefore, the PEDOT: PSS-sorbitol-maltitol can combine both advantages, as the the haze value is reduced to 0.21%, the $R_q$ is decreased to 0.93 nm, and the conductivity is improved, as shown in Table 3. In addition, it compares mixtures of sorbitol and maltitol in ratios of

1:1 and 1:3, as shown in Table 3. The $R_q$, haze, sheet resistance, and conductivity of these different mixture ratios are worse than those of a mixture of sorbitol and maltitol at a 3:1 ratio.

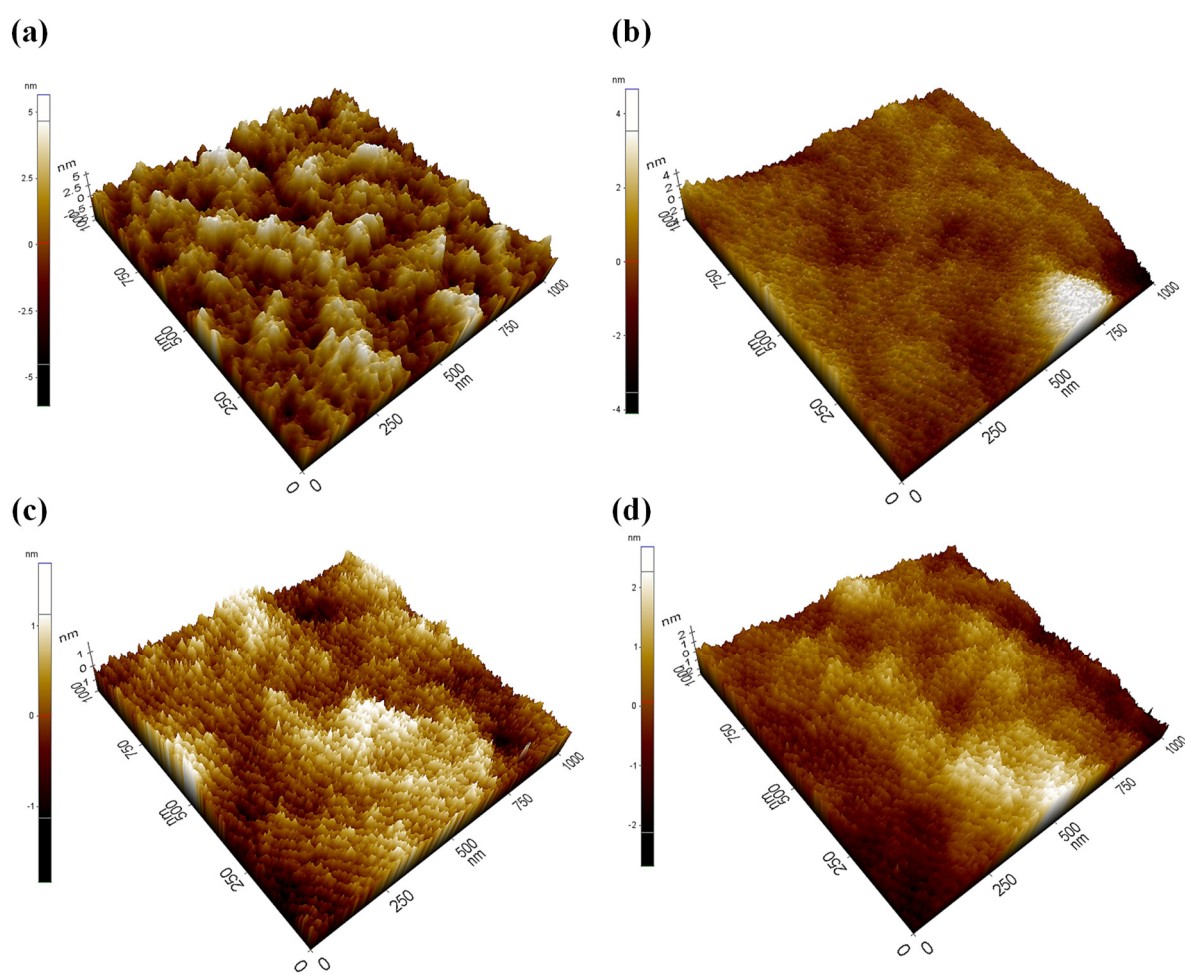

**(a)** **(b)** **(c)** **(d)**

**Figure 4.** Three-dimensional AFM images of PEDOT: PSS doped with different organic additives: (**a**) PEDOT: PSS, (**b**) PEDOT: PSS-sorbitol (4 wt%), (**c**) PEDOT: PSS-maltitol (4 wt%), (**d**) PEDOT: PSS-sorbitol-maltitol (4 wt%; 3:1).

**Table 3.** Transmittance, haze, sheet resistance, conductivity and $R_q$ of PEDOT: PSS without and with doped different organic additives.

| Sample | Transmittance (%) | Haze (%) | Sheet Resistance (Ω/sq) | Conductivity (S/m) | $R_q$ (nm) | Thickness (nm) |
|---|---|---|---|---|---|---|
| PEDOT: PSS | 87.5 | 0.31 | $6.2 \times 10^5$ | 6.3 | 1.21 | 152.9 |
| Sorbitol 4 wt% | 90.3 | 0.14 | $1.7 \times 10^5$ | 22.1 | 0.80 | 159.1 |
| Maltitol 4 wt% | 86.2 | 0.34 | $2.3 \times 10^5$ | 17.0 | 1.34 | 165.8 |
| Sorbitol: Maltitol 1:3 | 87.1 | 0.29 | $1.6 \times 10^5$ | 24.4 | 1.29 | 173.9 |
| Sorbitol: Maltitol 1:1 | 88.0 | 0.26 | $1.9 \times 10^5$ | 20.5 | 1.31 | 162.4 |
| Sorbitol: Maltitol 3:1 | 87.9 | 0.21 | $1.5 \times 10^5$ | 26.0 | 0.93 | 157.3 |

### 3.3. Scanning Electron Microscopy Analysis of PEDOT: PSS Doped with the Organic Additive

The surface morphology of PEDOT: PSS film without and with organic solution was measured by scanning electron microscopy (SEM), as shown in Figure 5. In Figure 5a, the raw polymer shows a uniform and smooth surface on the PEDOT: PSS film, without aggregates. In Figure 5b, the hydroxy sulfite bonds between negatively charged groups of

PSS and OH groups of sorbitol could promote the formation of protrusions which can also be considered as the dispersions of PEDOT: PSS micelles on the surface. It is illustrated that adding sorbitol causes the bond-breaking reaction, resulting in the dispersion of PEDOT: PSS micelles. The reaction can increase the distance of the interchain of PEDOT: PSS micelles. From the SEM images we can tell there is less difference between Figures 5a and 5c. It is conjectured that PEDOT: PSS-maltitol presents phase separation. In Figure 5d, there are small protrusions on the surface of the PEDOT: PSS-sorbitol-maltitol film. This is due to the fact that PEDOT: PSS-sorbitol-maltitol possesses both properties: the dispersion of PEDOT: PSS micelles and phase separation. A schematic of the phase separation is shown in Figure 6. In Figure 6, added maltitol in PEDOT: PSS is attributed to the decline of Coulomb interaction between PEDOT and PSS, resulting in the phase separation of PEDOT and PSS and forming a highly enriched PSS layer on the surface of the PEDOT: PSS film [27]. This phenomenon, phase separation, can enhance stability for the structure of PEDOT: PSS.

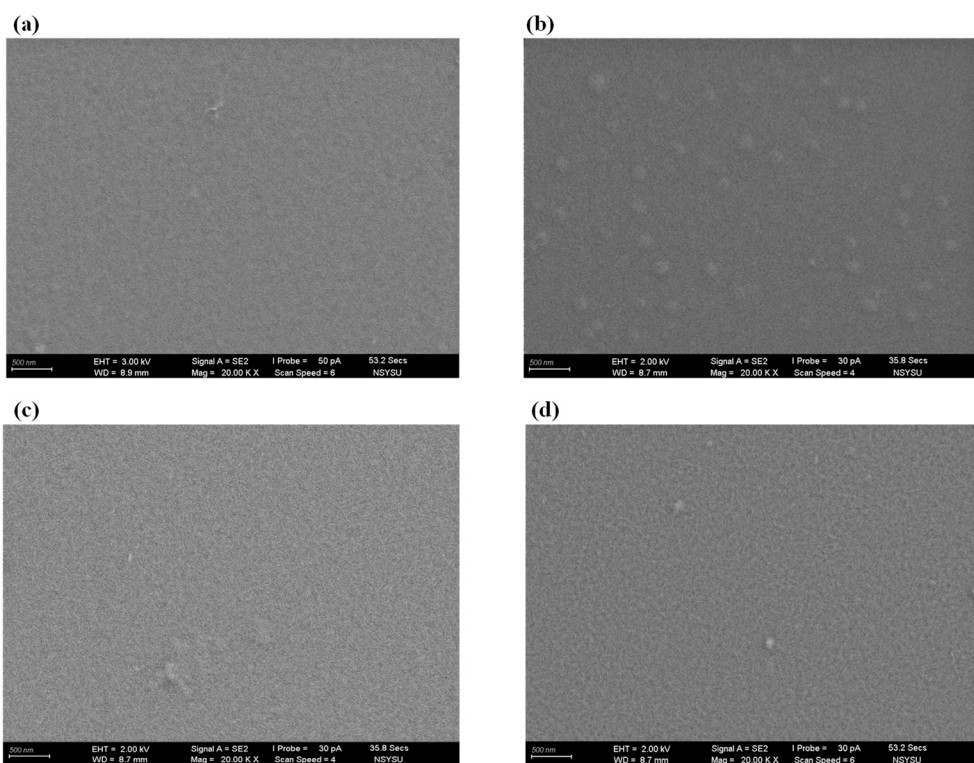

**Figure 5.** SEM images of (**a**) PEDOT: PSS, (**b**) PEDOT: PSS-sorbitol (4 wt%), (**c**) PEDOT: PSS-maltitol (4 wt%), (**d**) PEDOT: PSS-sorbitol-maltitol (3:1).

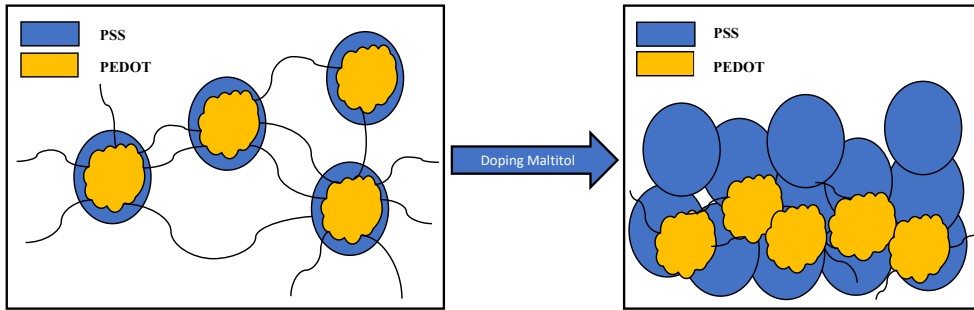

**Figure 6.** A schematic of maltitol added to PEDOT: PSS, presenting phase separation between PEDOT and PSS.

### 3.4. FTIR Analysis of PEDOT: PSS Doped with Organic Additives

To further understand the internal bonding of PEDOT: PSS doped with organic additives, the data for PEDOT: PSS without and with organic additives obtained from FTIR was normalized and is shown in Figure 7. All the spectra show the typical bands for PEDOT: PSS without and with organic additives from 4000 cm$^{-1}$ to 1000 cm$^{-1}$, as shown in Figure 7a. The band at 3269 cm$^{-1}$ is assigned to the O−H stretching of the organic additive, as shown in Figure 7b. It is worth noting that there is no change for intensity of the peak at 3269 cm$^{-1}$. This is because the excess hydrogen groups will react with hydrogen sulfite after adding sorbitol and maltitol, i.e., the bond-breaking reaction occurs. As shown in Figure 7c, the strong band at 2413 cm$^{-1}$ is contributed from the hydrogen bond between hydrogen sulfite, which is consistent with Figure 3b. PEDOT: PSS-sorbitol has fewer hydrogen bonds due to a greater reaction between PEDOT: PSS particles. The fewer hydrogen bonds show that there are more hydroxy group reactions with hydrogen sulfite. The greater the reaction between hydroxy groups and hydrogen sulfite is, the greater the dispersion of PEDOT: PSS particles is. This is why the transparency of PEDOT: PSS-sorbitol film is improved. As shown in Figure 7d, it is worthwhile to note that the bands at 1565 cm$^{-1}$ and 1478 cm$^{-1}$ are assigned to quinoid and benzenoid structures, respectively, in PEDOT: PSS [28]. The peak of 1565 cm$^{-1}$ is a coil-like PEDOT chain from the PEDOT: PSS film and the peak of 1478 cm$^{-1}$ is a linear PEDOT chain from the PEDOT: PSS-maltitol film [29]. The PEDOT chain transformation from the coil-like structure to the linear structure is due to the added maltitol in PEDOT: PSS. The appearance of peaks at 1220 cm$^{-1}$ and 1147 cm$^{-1}$ are ascribed to sulfite-containing groups (S=O) which are symmetric and asymmetric, respectively. The S=O is produced from sulfite ions. The sulfite ions are produced from breaking the bonds of hydrogen sulfite. Among the curves of Figure 7d, the intensity of the PEDOT: PSS-sorbitol curve (red line) is greater than that of the other curves. Therefore, the transparency of the PEDOT: PSS-sorbitol film is increased due to an increase in sulfite ions. In addition, the band at 1020 cm$^{-1}$ is contributed by C−O stretching in the maltitol additive.

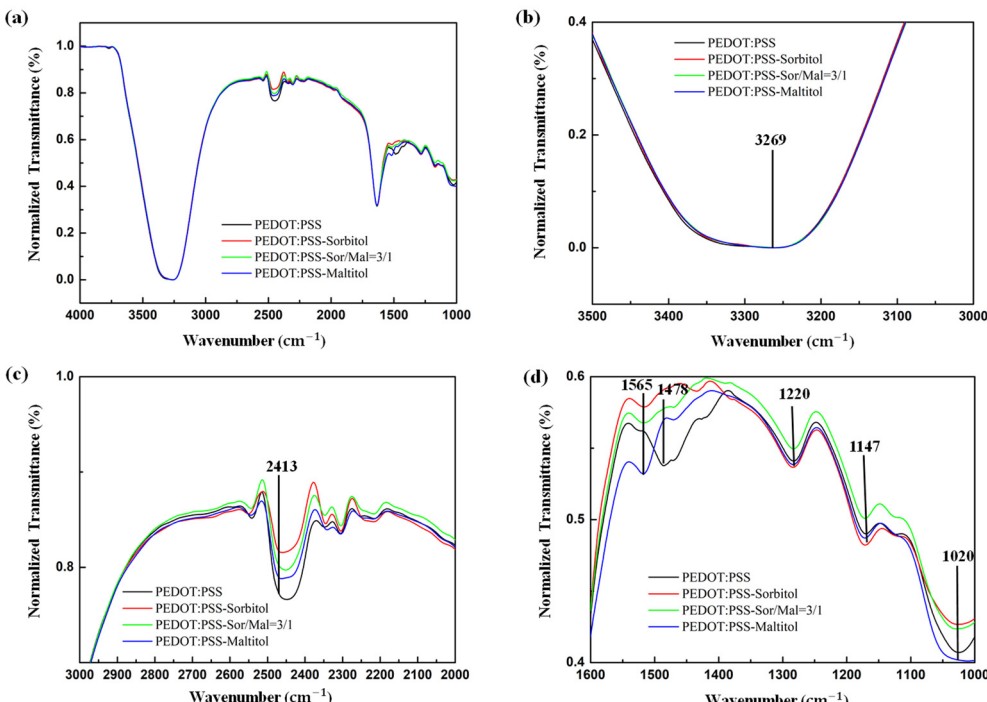

**Figure 7.** Normalized FTIR spectra of PEDOT: PSS without and with organic additives at (**a**) 4000~1000 cm$^{-1}$, (**b**) 3500~3000 cm$^{-1}$, (**c**) 3000~2000 cm$^{-1}$, and (**d**) 1600~1000 cm$^{-1}$.

### 3.5. Proton Nuclear Magnetic Resonance ($^1H$ NMR) Analysis of PEDOT: PSS Doped with Organic Additives

The data for PEDOT: PSS without and with organic additives obtained from $^1$H NMR is shown in Figure 8. The peak of 4.01 ppm represents the hydrogen bonds between PEDOT: PSS micelles, as shown in Figure 8a. The peak (4.01 ppm) of NMR disappears. This is due to the bond-breaking reaction caused by adding sorbitol to PEDOT: PSS, as shown in Figure 8b. The shift of the NMR peak from 4.01 ppm to 3.74 ppm is due to the mixture of sorbitol and maltitol in PEDOT: PSS. Mixing maltitol in PEDOT: PSS leads the film to have a lesser bond-breaking reaction, as shown in Figure 8c. In addition, the peak for PEDOT: PSS-maltitol shifts to 3.90 ppm, as shown in Figure 8d. In general, the formation of hydrogen bonds causes the peak to shift to a higher frequency (higher ppm) due to deshielding [30].

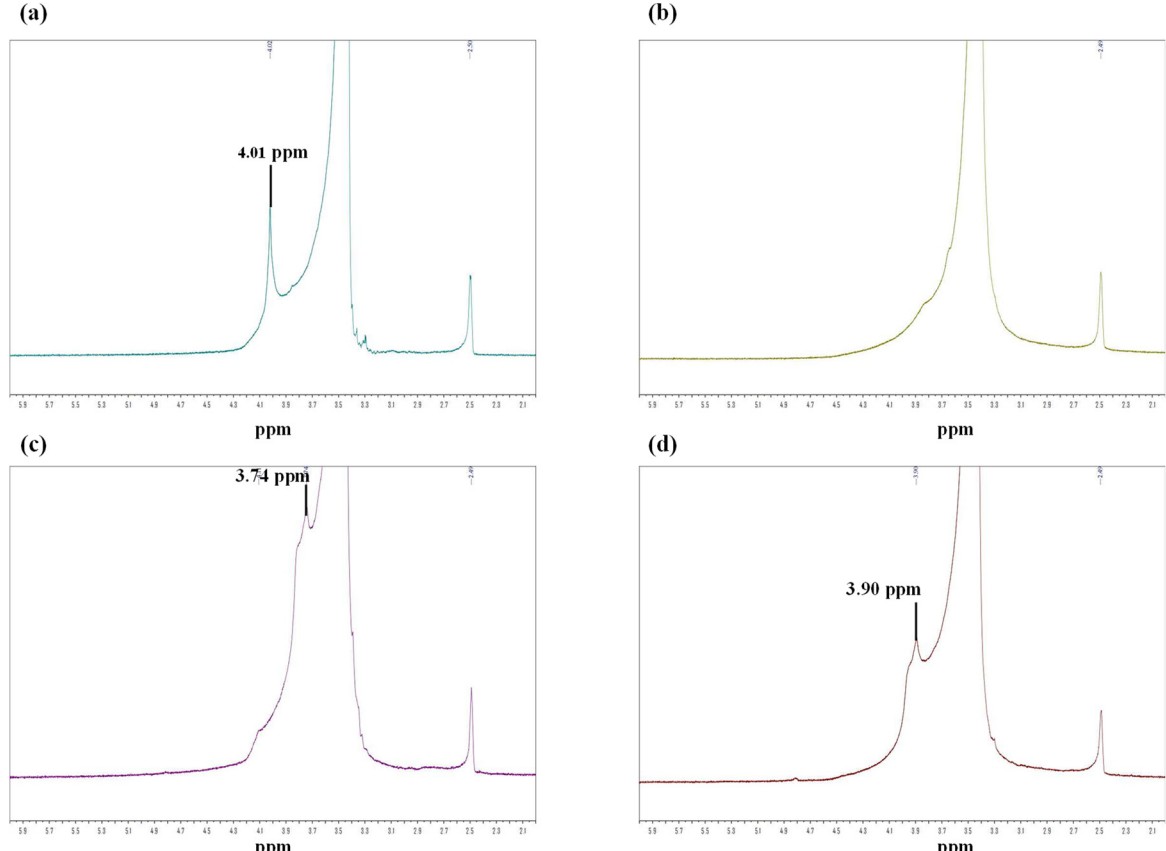

**Figure 8.** Nuclear magnetic resonance spectroscopy ($^1$H NMR) of (**a**) PEDOT: PSS, (**b**) PEDOT: PSS-sorbitol, (**c**) PEDOT: PSS-sorbitol-maltitol, (**d**) PEDOT: PSS-maltitol.

### 3.6. Differences between Acid Treatment and Multi-Acid Treatment in PEDOT: PSS Films

The low haze value of the PEDOT: PSS film can be optimized by changing the uniformity of the film and adding organic additives, but the conductivity cannot be effectively enhanced by the abovementioned methods. However, the conductivity of PEDOT: PSS film can be improved via FA treatment. Figure 9a shows a large amount of rough surface on the film because the PSS layer was removed by FA treatment, with the result that the thickness of the PEDOT: PSS film has declined from 152.9 nm to 135.7 nm. Although the roughness of the PEDOT: PSS film is increased from 1.21 nm to 1.26 nm, the haze value is reduced from 0.31% to 0.28%. It is confirmed that the FA treatment does not cause much damage to the evenness of the PEDOT: PSS film. After FA treatment, the reduction in the excess PSS layer around the PEDOT particles leads to these particles clumping together in the PEDOT: PSS film and allows the conductivity of the PEDOT: PSS film to increase [31]. In Figure 9b, there are some undulations on the surface of the PEDOT: PSS-sorbitol film. It

is assumed that the undulation on PEDOT: PSS-sorbitol film is due to the dispersion of PEDOT: PSS particles. After FA treatment, the PSS layer is removed by FA and the PEDOT particles are successively exposed. Hence the conductivity of PEDOT: PSS-sorbitol film enhances to $2.2 \times 10^4$ S/m. In Figure 9c, there is a sand-dune structure on the surface of the PEDOT: PSS-maltitol film. The growth mechanism is similar to that of PEDOT: PSS-sorbitol film after FA treatment. Therefore, the conductivity of PEDOT: PSS-maltitol film enhances to $1.6 \times 10^4$ S/m, as shown in Table 4. It is worth noting that the conductivity of PEDOT: PSS-maltitol is less than that of PEDOT: PSS-sorbitol. It is presumed that the FA treatment do not completely clean the PSS layer of the PEDOT: PSS-maltitol film due to phase separation. In Figure 9d, the surface of the PEDOT: PSS-sorbitol-maltitol film is more uniform than that of PEDOT: PSS-maltitol. This is attributed to the fact that sorbitol and maltitol mixture can not only disperse PEDOT particles on the surface of the film, but also cause phase separation after FA treatment. Thus, the PEDOT: PSS-sorbitol-maltitol film keeps a low haze value, low $R_q$, and low sheet resistance, as shown in Table 4. Based on above statement, the FA treatment is necessary to remove insulating PSS.

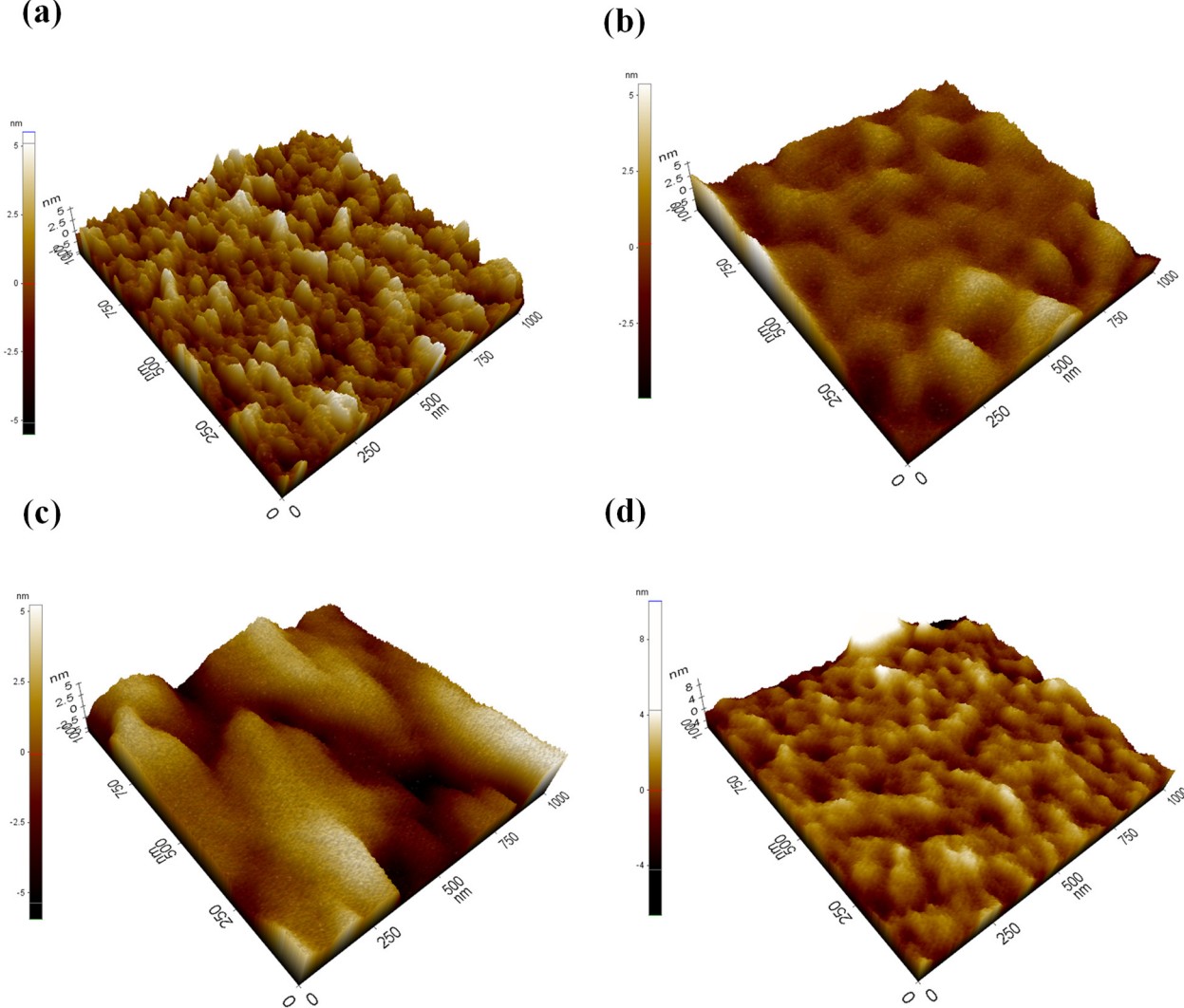

**Figure 9.** Three-dimensional AFM images of PEDOT: PSS without and with different organic additives and subjected to FA treatment: (**a**) PEDOT: PSS, (**b**) PEDOT: PSS-sorbitol (4 wt%), (**c**) PEDOT: PSS-maltitol (4 wt%), (**d**) PEDOT: PSS-sorbitol-maltitol (4 wt%; 3:1).

**Table 4.** Transmittance, haze, sheet resistance, conductivity, and $R_q$ of PEDOT: PSS without and with different organic additives and subjected to FA treatment (before and after).

| Sample | PEDOT: PSS | | Sorbitol 4 wt% | | Maltitol 4 wt% | | Sorbitol: Maltitol 3:1 | |
|---|---|---|---|---|---|---|---|---|
| | Before | After | Before | After | Before | After | Before | After |
| Transmittance (%) | 87.5 | 87.8 | 90.3 | 89.1 | 86.2 | 86.9 | 87.9 | 88.0 |
| Haze (%) | 0.31 | 0.28 | 0.14 | 0.17 | 0.34 | 0.41 | 0.21 | 0.25 |
| Sheet Resistance (Ω/sq) | $6.2 \times 10^5$ | 235 | $1.7 \times 10^5$ | 175 | $2.3 \times 10^5$ | 243 | $1.5 \times 10^5$ | 154 |
| Conductivity (S/m) | 6.3 | $1.6 \times 10^4$ | 22.1 | $2.2 \times 10^4$ | 17.0 | $1.6 \times 10^4$ | 26.0 | $2.5 \times 10^4$ |
| $R_q$ (nm) | 1.21 | 1.26 | 0.80 | 1.06 | 1.34 | 1.48 | 0.93 | 1.26 |
| Thickness (nm) | 152.9 | 135.7 | 159.1 | 142.4 | 165.8 | 141.3 | 173.9 | 144.5 |

Although FA treatment can reduce the value of sheet resistance, it is still not low enough (100 Ω/sq) for application in touchscreens. Hence, multi-FA treatment is an efficient way to remove insulating PSS, i.e., the roughness of film is increased after multi-FA treatment. Multi-FA treatment is chosen as the lesser of two evils to keep a low haze value and low sheet resistance. Figure 10a shows some unevenness on the surface of PEDOT: PSS film after multi-FA treatment because of the reduction in PSS, resulting in a decrease in the thickness of the film. Although the increasing haze value and $R_q$ are due to multi-FA treatment, the sheet resistance value is decreased nearly four-fold from 235 Ω/sq to 60 Ω/sq, as shown in Table 5. In addition, there is no datum for PEDOT: PSS-sorbitol due to the entire film peeling off after multi-FA treatment. This result shows that the adhesion of the PEDOT: PSS-sorbitol film is poor. In Figure 10b, there is a rough surface on the PEDOT: PSS-maltitol film after multi-FA treatment due to the stripping of residual PSS, resulting in a reduction in the thickness of the film. Thus, the $R_q$ is increased to 1.51 nm, but the sheet resistance of PEDOT: PSS-maltitol film is reduced to 170 Ω/sq. Despite multi-FA treatment having disadvantages such as surface roughness and the acid resistance of the film, PEDOT: PSS-sorbitol-maltitol after multi-FA treatment can compensate for these drawbacks. As shown in Figure 10c, there is a more uniform surface on the PEDOT: PSS-sorbitol-maltitol film than on that of the PEDOT: PSS-maltitol and PEDOT: PSS film. This phenomenon is due to the dispersion of PEDOT particles by mixing sorbitol and maltitol with PEDOT: PSS, reducing the haze value and sheet resistance to 0.31% and 91 Ω/sq, respectively. This result in our study is comparable with that of Hidenori Okuzaki et al. [18].

**(a)**          **(b)**

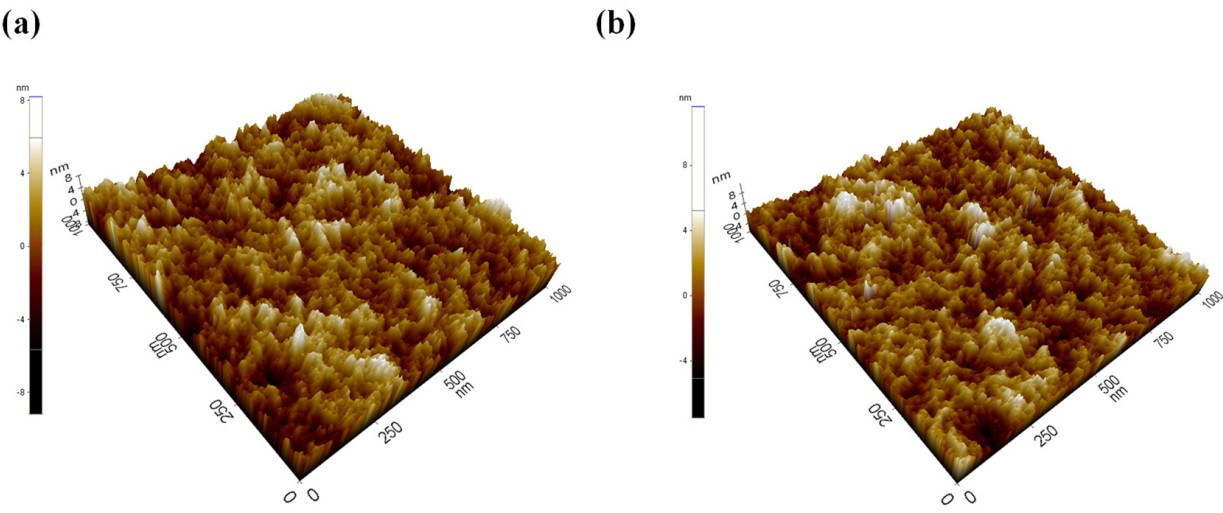

**Figure 10.** *Cont.*

**(c)**

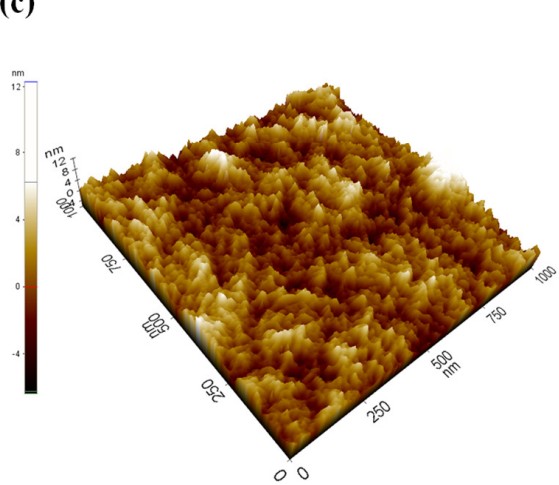

**Figure 10.** Three-dimensional AFM images of PEDOT: PSS without and with different organic additives and subjected to multi-FA treatment: (**a**) PEDOT: PSS, (**b**) PEDOT: PSS-maltitol (4 wt%), (**c**) PEDOT: PSS-sorbitol-maltitol (4 wt%; 3:1).

**Table 5.** Transmittance, haze, sheet resistance, conductivity, and $R_q$ of PEDOT: PSS without and with different organic additives and subjected to multi-FA treatment.

| Sample | Transmittance (%) | Haze (%) | Sheet Resistance (Ω/sq) | Conductivity (S/m) | $R_q$ (nm) | Thickness (nm) |
|---|---|---|---|---|---|---|
| PEDOT: PSS Sorbitol 4 wt% | 84.2 | 0.65 | 60 | $6.5 \times 10^4$ | 1.67 | 128.1 |
| Maltitol 4 wt% | 86.5 | 0.45 | 170 | $2.3 \times 10^4$ | 1.51 | 132.5 |
| Sorbitol: Maltitol 3:1 | 87.8 | 0.31 | 91 | $4.2 \times 10^4$ | 1.32 | 129.6 |

*3.7. Scanning Electron Microscopy Analysis of Acid Treatment and Multi-Acid Treatment in PEDOT: PSS Films*

The surface morphology of PEDOT: PSS film with acid treatment and multi-acid treatment was measured by SEM, as shown in Figure 11. In Figure 11a, the raw polymer shows a smooth PEDOT layer on the surface of the PEDOT: PSS film after acid treatment. In Figure 10b, there are some protrusions on the film, which are regarded as micelles (red circles) of PEDOT. From the SEM images we can tell that there is less difference between Figure 11a,c, illustrating that the PSS layer is removed by acid treatment. In Figure 11d, there are fewer protrusions on the surface of the PEDOT: PSS-sorbitol-maltitol film than that shown in Figure 11b. These protrusions are regarded as the micelles of PEDOT.

The surface morphology of the PEDOT: PSS film with multi-acid treatment was analyzed by SEM, as shown in Figure 12. In Figure 12a, the PEDOT layer is exposed on the surface of the film through multi-acid treatment. In Figure 12b, the dense film is revealed, illustrating that the phase separation can make the film denser. In Figure 12b, the surface of the PEDOT: PSS-sorbitol-maltitol film has unveiled some dispersed PEDOT micelles.

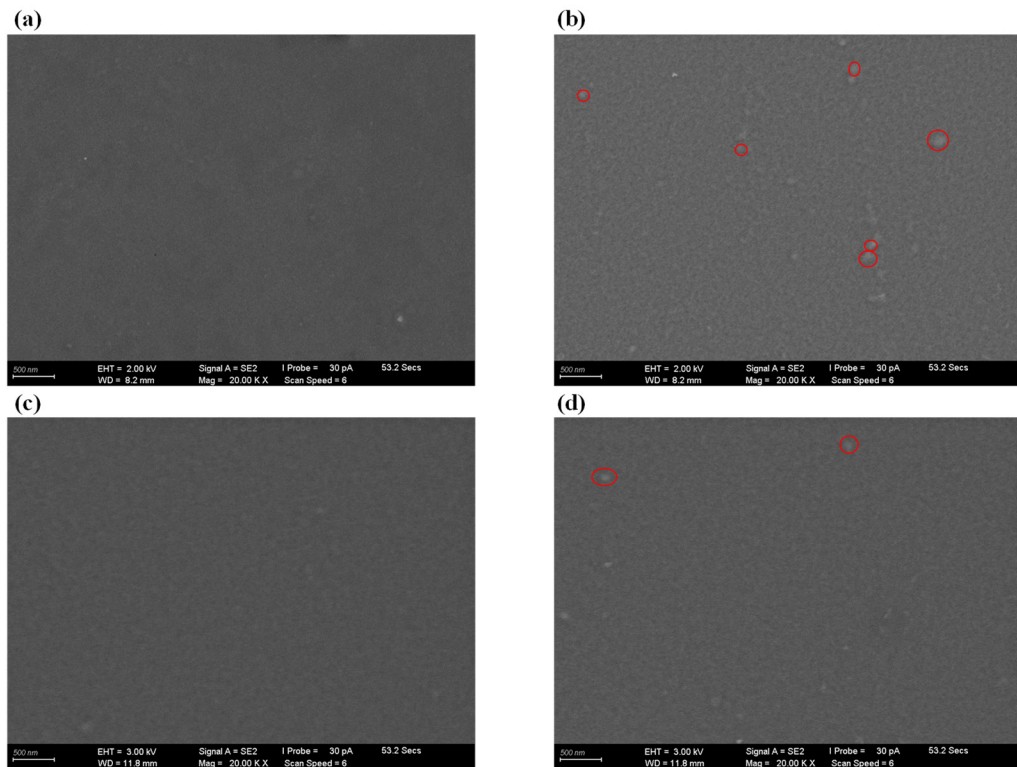

**Figure 11.** SEM images of PEDOT: PSS without and with different organic additives and subjected to FA treatment: (**a**) PEDOT: PSS, (**b**) PEDOT: PSS-sorbitol (4 wt%), (**c**) PEDOT: PSS-maltitol (4 wt%), (**d**) PEDOT: PSS-sorbitol-maltitol (3:1).

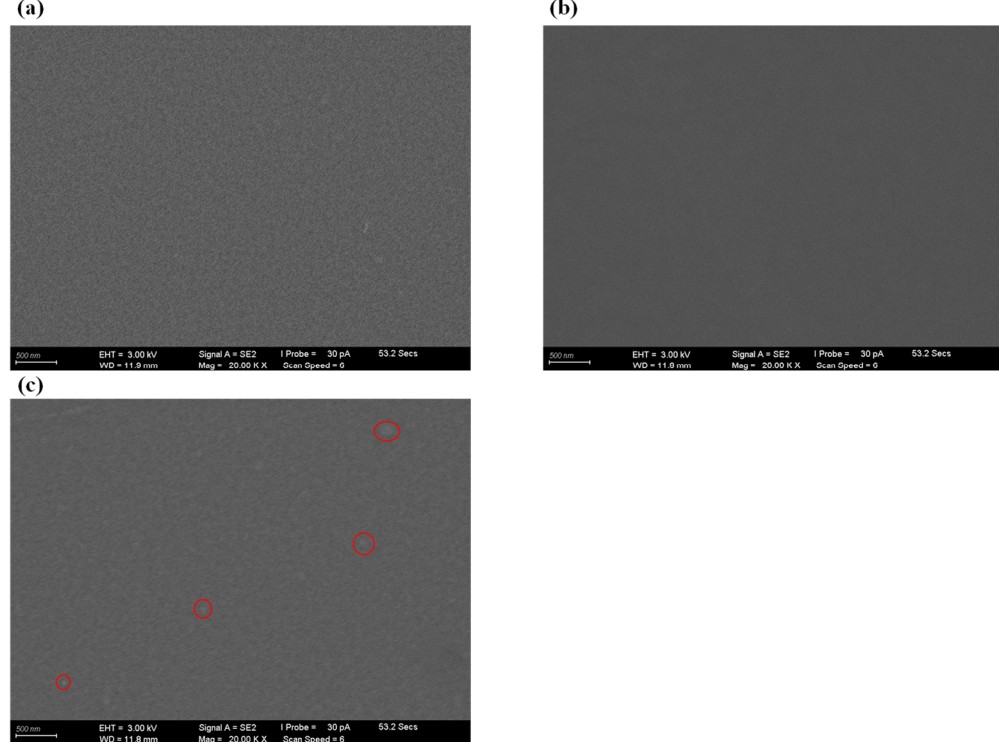

**Figure 12.** SEM images of PEDOT: PSS without and with different organic additives and subjected to multi-FA treatment: (**a**) PEDOT: PSS, (**b**) PEDOT: PSS-maltitol (4 wt%), (**c**) PEDOT: PSS-sorbitol-maltitol (3:1).

## 4. Conclusions

In summary, the PEDOT: PSS-sorbitol-maltitol undergoing multi-treatment can achieve a low haze value (0.31%) and low sheet resistance (91 Ω/sq). In our study, the mechanism of the bond-breaking reaction was investigated and a model for it was proposed. Based on the above, the haze of the PEDOT: PSS film was enhanced by adding organic additives. In addition, multi-FA treatment reduced the content of PSS anions, resulting in the increased conductivity of the polymer. Various techniques—four-point probe, haze meter, SEM, and AFM—show that the PEDOT: PSS-sorbitol-maltitol film is of good transparency and high conductivity.

**Author Contributions:** Conceptualization, S.-Y.L., P.-W.S., N.-F.W. and P.-C.L.; formal analysis, S.-Y.L., P.-C.L. and C.-J.H.; funding acquisition, P.-C.L. and S.-Y.L.; investigation, S.-Y.L. and C.-J.H.; resources, P.-C.L.; supervision, W.-R.C., S.-Y.L., C.-H.L., N.-F.W. and C.-J.H.; writing—original draft, P.-C.L. All authors have read and agreed to the published version of the manuscript.

**Funding:** This research was funded by the Ministry of Science and Technology (MOST) of the Republic of China: grant number 110-2221-E-390-019.

**Institutional Review Board Statement:** Not applicable.

**Informed Consent Statement:** Not applicable.

**Data Availability Statement:** The data presented in this study are available on request from the corresponding author.

**Conflicts of Interest:** The authors declare no conflict of interest.

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
