# Peer review of "Improving Optoelectrical Properties of PEDOT: PSS by Organic Additive and Acid Treatment"

_crystals, doi:10.3390/cryst12040537_

Round 1

Reviewer 1 Report

Lien, Lin Chen et al: “Improving Optoelectrical Properties of PEDOT: PSS by Organic Solvent and Acid Treatment”

The authors studied the effects of treatment with solvents and organic acids on the structural and optical properties of poly(3,4-ethylenedioxythiophene): polystyrene sulfonate (PEDOT: PSS; PP). Sorbitol and maltitol additives were found to improve the dispersion of PEDOT: PSS, and hence to reduce the haze of polymer films. Treatment with formic acid partially replaced the PSS content and increased the film conductivity.

Such an effect of low molar mass acids on PEDOT:PSS properties is not unknown, and it is similar to analogous effects observed in doped polyanilines (e.g. PAni:PSS, PAni:CSA, etc.)  Therefore this part of the authors’ work is not based on an original idea.  Also, there is previous work (some cited in this paper) on the effects of sorbitol on PEDOT:PSS dispersion; however, I’m not aware of any convincing previous study of the effects of maltitol as a dispersion enhancer, so this paper does appear to show some novel features. The authors need to emphasize the novel aspects and value of their work.

The paper is quite logically structured, but the English is not satisfactory and should be revised thoroughly, preferably with the help of editorial support or a fluent English speaker. There are some incorrect terms, and too many typographic, grammatical and orthographic errors.

e.g. in line 3 (and elsewhere):  “solvent” should be replaced by “dispersant” or “additive”. In any case, it is not realistic to refer to crystalline sugar compounds like sorbitol or maltitol as solvents.

Line 25:  “molecules” should probably be “micelles”, since there is strong evidence that conducting polymers like doped PEDOT and PAni are obtained as colloidal dispersions and not in true solution in any solvents.

Lines 26-27: “treatment with multiple formic acids can reduce content of the PSS, which increases the conductivity of PEDOT: PSS.” should be “multiple treatments with formic acid were found to reduce the PSS content of PEDOT:PSS and increase its conductivity.”

Line 29: “low haze for 0.3 % and low sheet resistance for 91 Ω/sq” should be “low haze of 0.3 % and low sheet resistance of 91 Ω/sq”.

line 40: “ploy (3,4-ethylene dioxythiophene)” should be “poly(3,4-ethylenedioxythiophene)”

lines 56-57: “the PP film is still high transparence” should be “the PP film was still highly transparent”

Line 252: insulting should be insulating.

Lines 291-293: “Combining these advantages, sorbitol and maltitol was mixed in the ratio of 3:1 can not only improve the haze of the film but also stable the structure of the film. In addition, FA treatment can reduce the content of negatively charged PSS, increasing the electrical conductivity of PP.” should be “Combining these advantages, sorbitol and maltitol mixed in the ratio of 3:1 not only reduced the haze of the film but also stabilized its structure. In addition, FA treatment reduced the content of PSS anions, increasing the polymer’s electrical conductivity.”

(…and so on, many similar points throughout the text of the article.)

The clarity of the Abstract should be improved, and unnecessary repetition removed, e.g. in lines  25-26 “…disperse the molecules of PEDOT: PSS, which decreases the haze of PEDOT: PSS.”

The Introduction is too brief. It should outline some more relevant recent research.

In the Experimental section, the chemical formulae, e.g. C6H14O6, should have the numbers in subscript.

In the Results section, the conductivity results are satisfactory and credible, but the degrees of uncertainty and reproducibility are not clear.

The infrared section is generally well-written.

The SEM photos are not very clear. If possible, clearer images should be provided. Certainly the scale bars should have more legible numbers on them.

A general question about PEDOT:PSS:  Is it realistic to speak of a possible application of this polymer in touch-screens?  The polythiophene family, including PEDOT, can achieve high conductivities when doped, but their mechanical durability and photostability are much less than those of ITO.

Reviewer 2 Report

Attached

Reviewer 3 Report

The authors undertook the ambitious task of improving the conductivity of PEDOT: PSS, which is quite a challenge. However, in my opinion, they did not propose any original technical or material solutions. Work on increasing the conductivity of PEDOT: PSS has been carried out for years and includes, apart from doping with e.g. graphene or carbon nanotubes, also the use of organic admixtures. Unfortunately, the authors do not discuss the doping history of PEDOT: PSS in the introduction.

It is well known that, PEDOT:PSS has high work function, high transparency and good conductivity and is most popular HTL layer in polymer solar cells, because it reduces the energy barrier between the HOMO of the active layer and the work function of the ITO electrode and smoothens the ITO surface after spin coating. However, various problems of PEDOT:PSS in organic solar cells and other devices influence on the decrease the long-term stability and performance of devices. For example, aggregate of particles in water solution of PEDOT:PSS can be defined as defects of the device induce degradation of solar cells. Moreover, degradation of ITO surface and as a consequence the instability of the ITO/PEDOT:PSS interface can be caused by strong acid character of PSS in PEDOT:PSS. Additionally, hydrophilic character of PEDOT:PSS influence on inconstant film morphology and electrical properties of polymer devices. Unfortunatly Authors not investigated electrochemical properties of modified PEDOT:PSS. I propose to see the works, e.g .: Polymers 2020, 12, 565; doi:10.3390/polym12030565. ACS Appl. Mater. Interfaces 2014, 6, 2067−2073.

For this reason, the Introduction is very poor and requires a complete rewrite.

Similarly, the abstract and summary do not represent the achievements received by the authors and should therefore be redrafted.

No information was found in the article why two organic admixtures were taken into account: one was aliphatic and the other was aromatic, and that with a different number of OH groups. This requires careful completion.

The title is not precise. The wording of organic solvent should be specified in detail. The idea of work is unclear. PEDOT: PSS is typically produced and sold as an aqueous or toluene solution. Since the authors purchased PEDOT: PSS, I don't understand why they don't analyze the effect of the solvent from PEDOT: PSS (water) on the properties. What is the role of FA in this system? Is it primary, secondary, etc. type of doping?

The issues related to non-covalent interactions and hydrogen bonds should be analyzed in great detail using the 1H NMR method.

There were no photos of the produced layers that could show the application potential of the proposed method of producing layers for opto-electronic devices.

Specifying PEDOT: PSS as PP is wrong and should be changed to PEDOT: PSS. In the polymer chemistry, PP is reserved for polypropylene.

I do not recommend the article at the present form.

Reviewer 4 Report

  • Some references seem to be wrong, for example in introduction, Leif et al. mentioned in the text and not in the reference list.
  • It should be mentioned that the high cost of ITO is due to shortage of it on the Earth.
  • The materials description should be corrected.
  • The procedure for PEDOT:PSS treatment should be described under one common title (not 5 separate titles)
  • Description of instrument details of the used equipment and experimental parameters used is not complete.
  • The results with maltitol shown in Table 1 are not consistent with sorbitol as claimed in the text.
  • A contradictory statement is given in the last sentence on page 5, "Both of these different mixtures did not have better...". 
  • The results after sorbitol addition shown in Table 3 are not in agreement with the SEM micrographs shown in Fig. 5.How can sorbitol addition decrease the haze and Rq if the surface morphology is more rough and inhomogeneous?
  • The intensity of the OH stretching bands after the different treatments is difficult to be compared with each other because the measurements are made separately from each other. The same applies for the band at 2413 cm-1. Rather, intensities between two bands in a spectrum can be compared with the same intensity ratio in an other spectrum or normalization against a common band can be made in order to be able to compare the intensities in two didfferent spectra.
  • References are missing from the IR peak interpretation.

Round 2

Reviewer 1 Report

The scientific points and the specific textual corrections that I exemplified previously have been addressed. One noticeable exception is that in line 40, which was not fully corrected (ploy should be poly).

The authors have not thoroughly revised the rest of their article, and there are still some ungrammatical and slightly unclear sentences in the text, e.g. Line 124 "It consisted with the results of Ling et al". 

Nevertheless, I think that the authors' meaning is now sufficiently clear for the article to be accepted in its current form. Any small errors such as the above-mentioned one in line 40 may be corrected in proof.

Reviewer 2 Report

Authors have revised the manuscript as per the suggestions and concerned raised by the reviewer. The article can now be accepted in its present form

Reviewer 3 Report

I see progress in the revised version of the paper and I proposed published paper as is.

Reviewer 4 Report

The English language should still be improved and corrected.